nanotechnology/environmental chemistry/green chemistry

gas separation, metal-organic framework, membrane engineering, chemistry

**Authors for correspondence:**
Magdalena Malankowska
e-mail: magnal@unizar.es
Joaquín Coronas
e-mail: coronas@unizar.es

This article has been edited by the Royal Society of Chemistry, including the commissioning, peer review process and editorial aspects up to the point of acceptance.

# Pre-combustion gas separation by ZIF-8-polybenzimidazole mixed matrix membranes in the form of hollow fibres—long-term experimental study

Adelaida Perea-Cachero[1,2], Miren Etxeberría-Benavides[3], Oana David[3], Adam Deacon[4], Timothy Johnson[5], Magdalena Malankowska[1,2], Carlos Téllez[1,2] and Joaquín Coronas[1,2]

[1]Instituto de Nanociencia y Materiales de Aragón (INMA), CSIC-Universidad de Zaragoza, Zaragoza 50018, Spain
[2]Chemical and Environmental Engineering Department, Universidad de Zaragoza, Zaragoza 50018, Spain
[3]TECNALIA, Basque Research and Technology Alliance (BRTA), Energy and Environment Division, Membrane Technology and Process Intensification Group, Mikeletegi Pasealekua 2, Donostia-San Sebastián 20009, Spain
[4]Johnson Matthey Technology Centre, Process Chemistry and Catalysis Group, Chilton Site, Belasis Avenue, Billingham Cleveland TS23 1LB, UK
[5]Johnson Matthey Technology Centre, Recycling Technologies Group, Blount's Court, Sonning Common, Reading RG4 9NH, UK

AP-C, 0000-0002-0157-9507; ME-B, 0000-0002-5419-1573;
OD, 0000-0002-9586-5677; AD, 0000-0002-3110-7770;
TJ, 0000-0003-4651-7374; MM, 0000-0001-9595-0831;
CT, 0000-0002-4954-1188; JC, 0000-0003-1512-4500

Polybenzimidazole (PBI) is a promising and suitable membrane polymer for the separation of the $H_2/CO_2$ pre-combustion gas mixture due to its high performance in terms of chemical and thermal stability and intrinsic $H_2/CO_2$ selectivity. However, there is a lack of long-term separation studies with this polymer, particularly when it is conformed as hollow fibre membrane. This work reports the continuous measurement of the $H_2/CO_2$ separation properties of PBI hollow fibres, prepared as mixed matrix membranes with metal-organic framework (MOF) ZIF-8 as filler. To enhance the scope of the experimental approach, ZIF-8 was synthesized from the transformation of ZIF-L upon up-scaling the MOF synthesis

into a 1 kg batch. The effects of membrane healing with poly(dimethylsiloxane), to avoid cracks and non-selective gaps, and operation conditions (use of sweep gas or not) were also examined at 200°C during approximately 51 days. In these conditions, for all the membrane samples studied, the $H_2$ permeance was in the 22–47 GPU range corresponding to 22–32 $H_2/CO_2$ selectivity values. Finally, this work continues our previous report on this type of application (Etxeberria-Benavides *et al.* 2020 *Sep. Purif. Technol.* **237**, 116347 (doi:10.1016/j.seppur.2019.116347)) with important novelties dealing with the use of ZIF-8 for the mixed matrix membrane coming from a green methodology, the long-term gas separation testing for more than 50 days and the study on the membrane operation under more realistic conditions (e.g. without the use of sweep gas).

# 1. Introduction

Earth absorbs some of the radiant energy that is received from the Sun and either reflects some of it as light or radiates the rest back to space in the form of heat. The global temperature of Earth's surface depends on the balance between incoming and outgoing energy. If, for some reason, this energy balance is shifted, Earth's surface becomes warmer or cooler, leading to a variety of changes in a global climate. A variety of natural and man-made mechanisms can affect the global energy balance and force changes in Earth's climate. Since the beginning of the Industrial Revolution (around 1750) human activities have produced a 40% increase in the atmospheric concentration of $CO_2$ from 280 ppm in 1750 to 406 ppm in early 2017 [1]. There is an urgent need in decreasing the emission of $CO_2$ as well as in developing effective $CO_2$ capturing methods [2]. The so-called 'conventional solvents' for $CO_2$ absorption, which are well-known solvents that present high $CO_2$ absorption capacity, were commercialized in the 1930s. There are two main types of conventional solvents: (i) physical and (ii) chemical ones. The main characteristic of physical solvents is the need of high pressure to be efficient absorbers, otherwise, they should be replaced by chemical absorbers, for example, amines. An aqueous amine solution is used to capture $CO_2$ in the zwitterion reaction where $CO_2$ reacts with amine in the form of carbamate. Even though amines are very efficient in $CO_2$ capture mainly due to high reactivity and selectivity as well as high absorbing capacity, there are some serious and dangerous disadvantages: (i) high vapour pressure, (ii) emission of toxic compounds with high potential to pollute water due to their solubility in water and (iii) high desorption and recycling costs due to elevated reaction heat to name a few [2,3].

It is clear that alternative technologies for more efficient $CO_2$ separation need to be implemented. Membrane-based operations show a promising potential to replace the conventional energy-intensive technologies and provide reliable solutions for sustainable and ecological growth [4]. Membrane separation processes provide more surface area per unit volume than, for example, conventional packed towers, leading to higher mass transfer rates [5]. Thin-film technologies are one of the most studied techniques for the separation of $CO_2$ from non-polar gases, such as $H_2/CO_2$, $CO_2/N_2$ and $CO_2/CH_4$ gas mixtures, due to their significant advantages over conventional methods, i.e. low energy consumption, small footprint, mechanical simplicity and easy to scale up, to name a few [6]. Pre-combustion, which is one of the technologies from carbon capture and storage systems, is widely applied in power production, gaseous fuel and fertilizer fabrication. Briefly, after the fossil fuel oxidation, the CO from the resulting syngas (CO and $H_2$) is shifted into $CO_2$ and $H_2$ after reacting with steam ($H_2O$). The $CO_2$ can be captured, while $H_2$ can be used as a fuel because $CO_2$ is separated before the combustion occurs [7]. Gas mixtures can be efficiently separated by synthetic membranes made of, for example, polymers. Polybenzimidazole (PBI) is a promising candidate for the separation of a pre-combustion gas mixture ($H_2/CO_2$) due to its good chemical resistance, high thermal stability, high intrinsic $H_2/CO_2$ selectivity as well as impressive compression strength [8–10]. Besides, recently, the green processability of PBI in ethanol has been demonstrated to produce highly selective coatings of this polymer both on flat and hollow fibre (HF) polymeric supports [11].

In order to further improve the selective performance of a polymeric membrane, the preparation of mixed matrix membranes (MMMs) could be a possible solution and it is currently a topic of growing interest. MMMs are composed of a continuous polymeric matrix and a dispersed inorganic filler. Metal-organic frameworks (MOFs) are a growing class of organic–inorganic crystalline and porous materials. Because of their hybrid nature, MOFs are commonly used as fillers in MMMs due to the formation of good interaction with the polymers, avoiding the creation of non-selective gaps or cracks that are common when applying different types of fillers [12,13]. Zeolitic imidazolate frameworks (ZIFs) are a class of MOFs that exhibit a zeolite type structure, ZIF-8, with the SOD type topology, being the most well-known and studied ZIF. ZIF-8 is composed of the metal cation $Zn^{2+}$ linked to the 2-methylimidazolate ligand species. This results in large cavities of 1.16 nm interconnected through

windows of about 0.34 nm [14,15]. Moreover, gas separation membranes in the configuration of HFs are attractive, especially due to the possibility of a process intensification in terms of high surface to volume ratio, and thus an increased efficiency and favourable economy.

The goal of this work was to fabricate high-performance hollow fibre MMMs based on PBI with ZIF-8 as a filler and study their gas separation ability in the long term, mimicking the real pre-combustion environment considering the temperature, pressure and time. Mini membrane modules with 1 or 2 HFs were prepared and used for the continuous measurements during approximately 51 days. The effect of the healing with elastomer, to avoid cracks and non-selective gaps, was also examined. Moreover, ZIF-8 was synthesized from the modified method of transformation of ZIF-L in order to scale up the synthesis into a 1 kg batch.

# 2. Material and methods

## 2.1. Materials

For the synthesis of ZIF-8, 2-methylimidazole (Hmim) (99% purity) and zinc nitrate hexahydrate (99+% purity) were purchased from Alfa Aesar and absolute ethanol (99.8+% purity) was purchased from Fisher Scientific. For the fabrication of the HF membranes, Fumion AP poly(2,2′-(m-phenylene))-5,5′-bisbenzimidazole (PBI, $M_w = 48\,000\,\mathrm{g\,mol^{-1}}$) was supplied by Fumatech BWT GmbH. Polyvinylpyrrolidone (PVP) K30, anhydrous N-methylpyrrolidone (NMP, 99.5% purity) and lithium chloride (LiCl) were purchased from Sigma-Aldrich. n-Hexane and methanol were purchased from Fisher Scientific.

## 2.2. Methods

### 2.2.1. ZIF-8 synthesis

ZIF-8 was synthesized via the transformation of ZIF-L in ethanol—a modified preparation route established in a previous report [16]. ZIF-L was first synthesized by preparing a solution of Hmim in deionized water in the following molar ratio (1 : 34.7). In parallel, a separate solution of zinc nitrate was prepared in deionized water in the following molar ratio (1 : 8.75). The zinc solution was added at once to the imidazole solution, precipitation occurred immediately and the resulting mixture was stirred for 30 min. The obtained solid (ZIF-L) was collected by vacuum filtration, washed 3 times with deionized water and dried in a vacuum oven at 80°C overnight. The dried ZIF-L material was ground and sieved to a maximum diameter of 100 µm. The ZIF-L powder was transformed into ZIF-8 by dispersing it in absolute ethanol in the following molar ratio (1 : 57.7). The resulting mixture was stirred and refluxed for 72 h. Once complete, the reaction was left to cool to room temperature before separation by centrifuging at 4000 r.p.m. for 10 min. The resulting 1 kg solid was washed 3 times with absolute ethanol. The product was dried in ambient air for 16 h and then treated under vacuum at 80°C for 16 h [17].

### 2.2.2. Mixed matrix hollow fibre membrane fabrication

The fabrication of PBI mixed matrix HF membranes (with 182 µm and 275 µm inner and outer diameter, respectively) was based on a process of dry jet wet spinning followed by a wet quench. A PBI/PVP blend was used as polymer matrix and ZIF-8 as filler material at 10 wt% loading. The ZIF-8 loading was calculated as follows: $\mathrm{wt_{ZIF-8}}/(\mathrm{wt_{ZIF-8}} + \mathrm{wt_{PBI}}) \times 100$. For the spinning dope preparation, ZIF-8 was first dispersed in NMP by stirring overnight at room temperature. In parallel, PBI and PVP were dissolved in NMP (PBI/PVP weight ratio of 88/12) by stirring overnight at 80°C. PBI/PVP solution was then poured in the ZIF-8 dispersion and mixed until a homogeneous spinning dope solution was obtained. The HF preparation procedure has been described in detail elsewhere [10] using the spinning parameters detailed in table 1.

### 2.2.3. Hollow fibre membrane module preparation

The purpose of this work was to perform long-term gas separation measurements on the PBI HFs to mimic the industrial application of a pre-combustion process. Figure 1 shows the type of tests conducted in this work. First, the bundle of HFs was prepared by a method described above (see table 2 and electronic supplementary material, table S1, for the HF characteristics). The HF was placed in a stainless-steel mini module and sealed with a bicomponent epoxy resin (Araldite) at both ends. The resin was cured at room

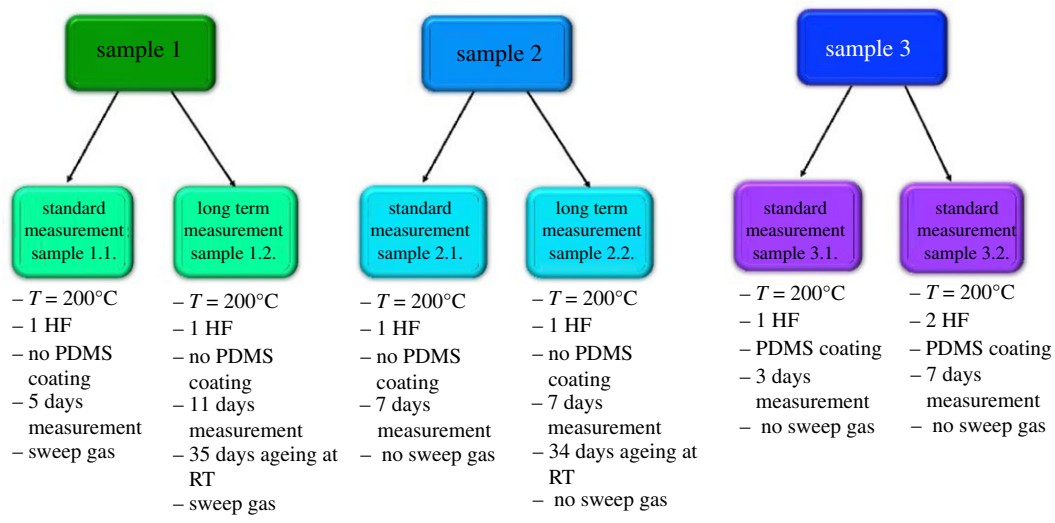

**Figure 1.** Graphical representation of the measurements conducted in this work. Samples 1, 2 and 3.1 corresponding to 1 HF mini module and sample 3.2 to 2 HFs mini module.

**Table 1.** Spinning parameters used for PBI mixed matrix hollow fibre membrane fabrication.

| spinning parameter | studied range |
| --- | --- |
| polymer concentration | 17 wt% |
| ZIF-8 loading | 10 wt% |
| bore composition | 50 wt% NMP/50 wt% $H_2O$ |
| spinneret temperature | 25°C |
| outer dope flow rate | 140 ml h$^{-1}$ |
| bore flow rate | 70 ml h$^{-1}$ |
| air gap height | 5 cm |
| quench bath temperature | 25°C |
| take up rate | 25 m min$^{-1}$ |

**Table 2.** Characteristics of fabricated mixed matrix hollow fibre membranes.

| fibre parameter | parameter range |
| --- | --- |
| fibre length | ≈10 cm |
| membrane thickness | ≈90 μm |
| fibre outer diameter | 275 ± 1 μm |
| fibre inner diameter | 182 ± 5 μm |

temperature for 24 h and the HF was then cut to a desired length. The mini module was handmade with 1/4 in tubing and 1/4 in tube fittings in the main body (where HFs were located). To fix the mini module to the gas separation system, quick couplers were mounted at the entries (feed and sweep gas) and exits (permeate and retentate) of the module through tubing and tube fittings of 1/8 in. When no sweep gas experiments were performed, the sweep gas entry was capped with a 1/4 in cap.

The measurements were then divided into two types: standard and long-term analyses. Long-term measurements were characterized by a long ageing process (34–35 days) at room temperature before the actual tests, while the standard measurement did not include the ageing process and were tested immediately after receiving. Previously to standard measurement, sample 2 was evacuated at 180°C under vacuum for 20 h. Moreover, some of the fibres (i.e. samples 3.1 and 3.2) were coated with a very

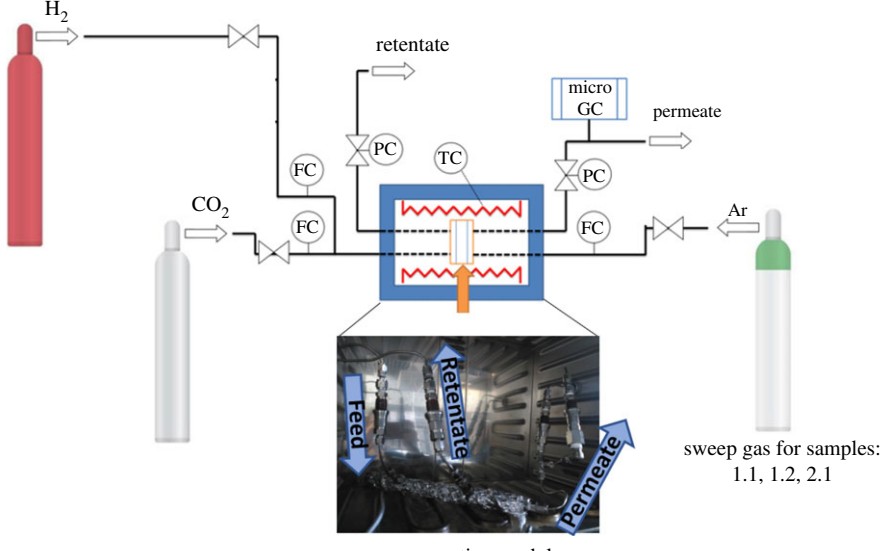

**Figure 2.** Gas permeation experimental system with indicated: inlet gas mixture ($H_2$ and $CO_2$), gas chromatograph (GC), sweep gas (whenever used) and the oven with inserted mini module.

thin layer of poly(dimethylsiloxane) (PDMS) (3 wt% in hexane) to obtain a healing effect and remove possible non-selective gaps or cracks on the selective layer of the membrane. The fibres were covered with PDMS for 30 s twice, allowing the solvent to evaporate at 40°C after each coating. These samples were finally activated at 150°C under vacuum for 24 h. Sample 1 was not post-treated with PDMS and was thus measured as received. Moreover, samples 2.2, 3.1 and 3.2 were measured without a sweep gas.

### 2.2.4. Gas permeation measurements

The $H_2/CO_2$ separation was performed in the experimental system shown in figure 2. All the measurements were carried out at 200°C. To prevent sealant degradation at such a high temperature, the mini module was wrapped by a glass fibre heating cable (SAF Wärmetechnik, KM-HC-G) without covering the sealed ends of the module to avoid their damage. A thermocouple was fixed in contact to the middle part of the mini module and was also wrapped by the heating cable to monitor the temperature. The permeation module was placed in a UNE 200 Memmert oven for protection. Gas separation measurements were carried out by feeding the pre-combustion gaseous mixture of $H_2/CO_2$ (25/25 $cm^3$ (STP) $min^{-1}$) at an operating pressure of 4 bar to the feed side (outer side of the fibre), controlled by two mass flow controllers (Alicat Scientific, MC-100CCM-D). Gas separation through samples 1.1, 1.2 and 2.1 was performed using sweep gas. By contrast, samples 2.2, 3.1 and 3.2 were measured without sweep gas to compare the results obtained by both methods. The membrane areas were 0.76–0.85 $cm^2$ (samples 1.1, 1.2, 2.1, 2.2 and 3.1) and 1.63 $cm^2$ (sample 3.2) for modules with 1 and 2 HFs, respectively (see electronic supplementary material, table S1). The permeate side of the membrane (inner side of the fibre) was thus swept or diluted, respectively, with 4.5 $cm^3$ (STP) $min^{-1}$ of Ar, at atmospheric pressure (approx. 1 bar) (Alicat Scientific, MC-5CCM-D). Concentrations of $H_2$ and $CO_2$ in the outgoing streams were analysed online by an Agilent 3000 A micro-gas chromatograph. Permeance was calculated in GPU ($10^{-6}$ $cm^3$ (STP) $cm^{-2}$ $s^{-1}$ $cm$ $Hg^{-1}$) once the steady state of the exit stream was reached (at least after 3 h). The separation selectivity was calculated as the ratio of permeances. Table 3 shows the parameters of the gas separation measurements.

## 2.3. Characterization methods

### 2.3.1. ZIF-8 characterization

Powder X-ray diffraction (PXRD) data were collected in reflection geometry using a Bruker D2 with Cu K$\alpha$ radiation ($\lambda = 1.5406$ Å) over the $5 < 2\theta < 50°$ range in 0.02° steps. Phase identification was conducted by comparison to simulated powder patterns. Thermogravimetric analysis (TGA) was conducted on a TA Instruments SDT Q600 or Q650 TGA system. The samples were held in an alumina pan, and the

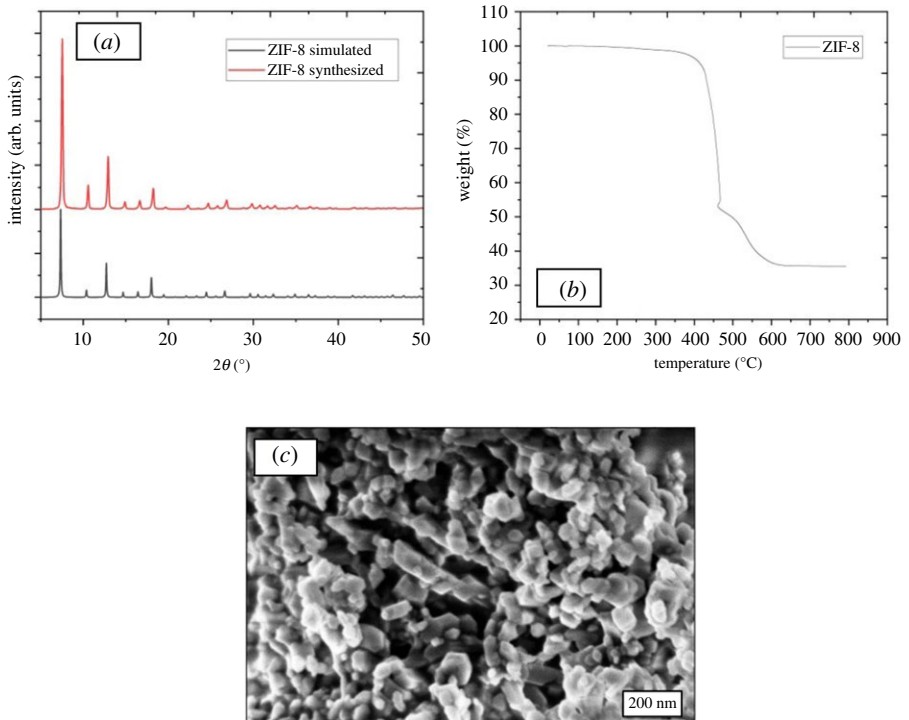

**Figure 3.** (*a*) PXRD pattern of synthesized ZIF-8 comparing to a simulated pattern of ZIF-8. (*b*) TGA curve of synthesized ZIF-8 measured in air. (*c*) SEM image of synthesized ZIF-8.

temperature was increased from room temperature to 800°C at a heating rate of 10°C min$^{-1}$ under a

**Table 3.** Gas separation measurement parameters.

| parameter | value |
|---|---|
| $\Delta P$ | 3 bar |
| $T$ | 200°C |
| $Q_{H2}$ | 25 cm$^3$(STP) min$^{-1}$ |
| $Q_{CO2}$ | 25 cm$^3$(STP) min$^{-1}$ |
| $Q_{Ar}$ | 4.5 cm$^3$(STP) min$^{-1}$ |
| stage cut | <1% |

continuous air flow was 100 ml min$^{-1}$. $N_2$ physisorption measurements were collected on a Micromeritics 3500 3Flex instrument. The samples were first degassed on a Micromeritics SmartVacPrep sample preparation device at 90°C under vacuum (10$^{-5}$ mm Hg) for 8 h followed by *in situ* of samples on the 3Flex instrument for 4 h at 90°C under vacuum (5 × 10$^{-5}$ mm Hg). The nitrogen sorption isotherms of degassed samples were recorded at liquid nitrogen temperature (77 K). Isotherms were subject to BET analysis for surface area calculation. Nanocrystal morphology and size were determined by scanning electron microscopy (SEM) using a Zeiss Ultra 55 field emission electron microscope equipped with in-lens secondary electron and backscattered detectors. High-resolution low-accelerating voltage imaging was acquired at accelerating voltage 1.6–5 kV, aperture 20–30 µm and working distance 3–4 mm.

### 2.3.2. Hollow fibre membrane characterization

SEM images of the HF membranes were obtained using a Quanta 250 ESEM (FEI) Inspect F50 model scanning electron microscope, operated at 10 kV as well as an Inspect F50 combined with energy-dispersive X-ray spectroscopy (EDX) operated at 20 kV. Cross sections of membranes were prepared by freeze-fracturing after immersion in liquid $N_2$, and subsequently coated with Pt.

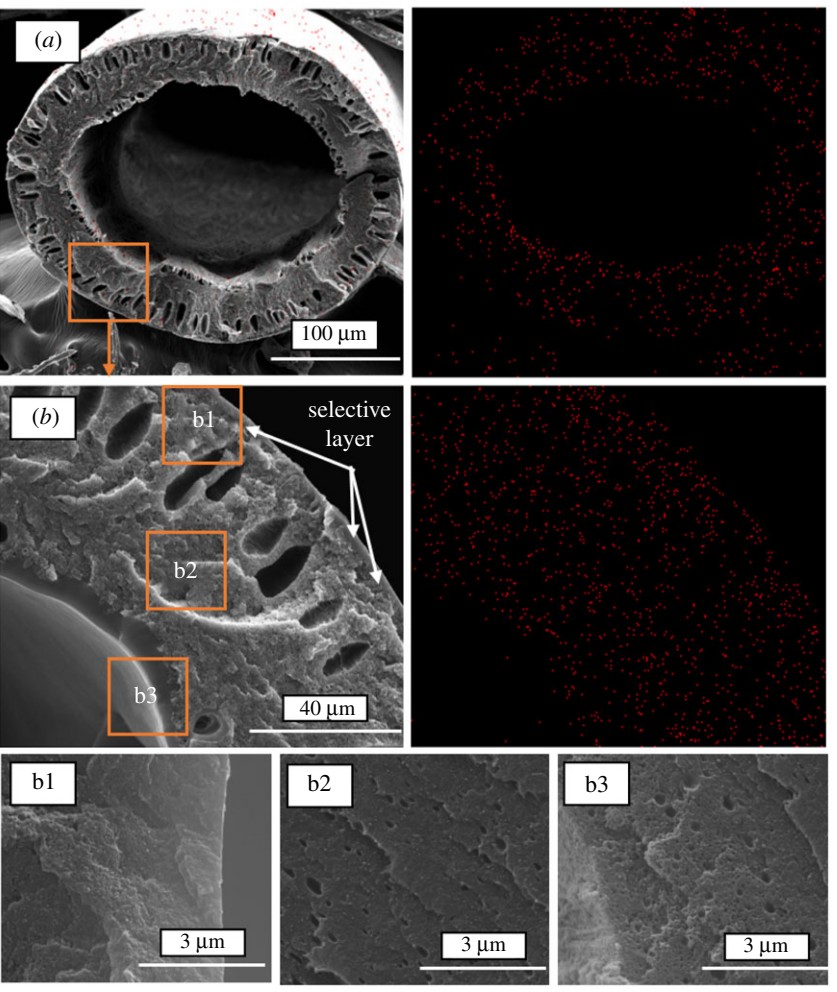

**Figure 4.** SEM images of the (*a*) overall fibre cross-section inlet and (*b*) the selective layer detail with their corresponding EDX images with red dots indicating Zn of ZIF-8. The images correspond to sample 3 with visible (especially in *b*) coating with PDMS.

# 3. Results and discussion

## 3.1. ZIF-8 characterization

The PXRD pattern for the synthesized ZIF-8 is consistent with the corresponding simulated pattern [18] indicating that a phase pure material has been produced (figure 3*a*), in agreement with the complete transformation of ZIF-L to ZIF-8. The simulated pattern was visualized using Mercury software from the Cambridge Crystallographic Data Centre [19].

The TGA trace, shown in figure 3*b*, is consistent with previous reports [15] and no degradation is seen until approximately 400°C. After this temperature, ZIF-8 degrades into a residue related to zinc oxide. $N_2$ physisorption measurements and the subsequent calculated BET specific surface area, equal to 1552 $m^2\,g^{-1}$, are consistent with ZIF-8 demonstrating no residual linker remains within the pores of the material. Finally, SEM image of ZIF-8 (figure 3*c*) shows the average particle size of approximately 100 nm.

## 3.2. Membrane characterization

SEM images of the HF cross sections are shown in figure 4. Fibres possess an outer diameter of approximately 275 µm and an inner diameter of approximately 185 µm. They present a cross section with a sponge-like substructure in the inner side of the fibre and an outermost part with the presence of few finger-like macrovoids, typically produced during the phase inversion. Since these are single layer type fibres, ZIF-8 is present in the entire fibre (the selective layer and the porous substructure) as

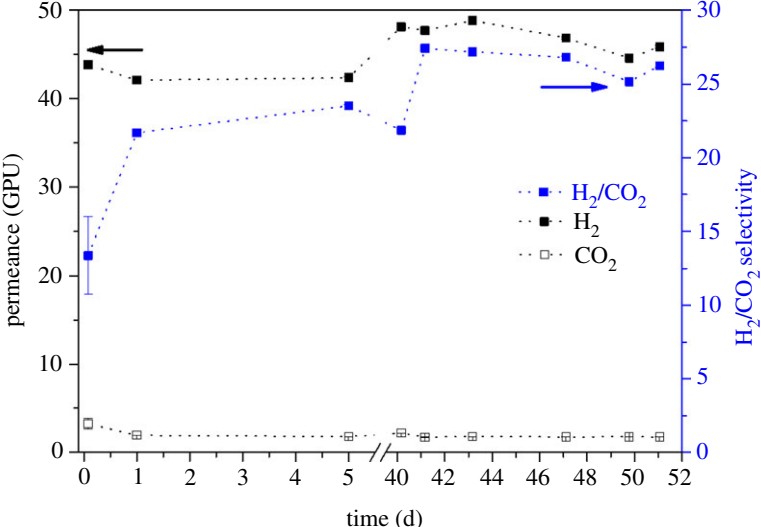

**Figure 5.** Long-term performance of HF membranes at 200°C and 4 bar feed pressure from samples 1.1 and 1.2 considering the $CO_2$ and $H_2$ permeances as well as $H_2/CO_2$ selectivity, with sweep gas.

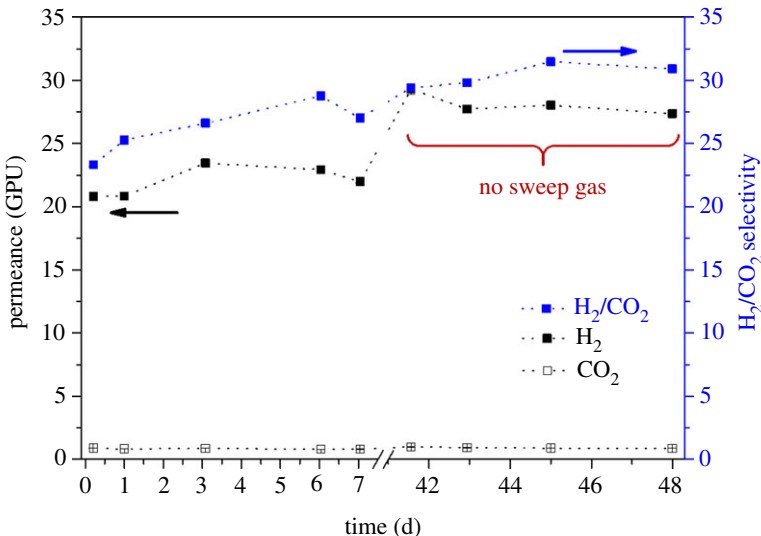

**Figure 6.** Long-term performance of HF membranes at 200°C and 4 bar feed pressure from samples 2.1 and 2.2 considering the $CO_2$ and $H_2$ permeances as well as $H_2/CO_2$ selectivity. Sweep gas used for sample 2.1 and no sweep gas used for sample 2.2 (indicated in the figure).

as can be seen from the EDX measurements (figure 4a,b). Nevertheless, as was mentioned in the description of the gas separation measurement, the gas mixture was fed from the outside of the fibre to the inner part, considering the outer part being the selective layer with a thickness about 3–4 µm.

## 3.3. Gas permeation measurements

As was demonstrated in previous work [10], the incorporation of ZIF-8 into the PBI membrane improved especially the mixed gas transport and it was expected that a higher temperature will positively influence the $H_2/CO_2$ separation. To demonstrate the reliability of the preparation method as well the final performance of the membranes, two different batches of the same type of HF (samples 1 and 2) were measured. Moreover, the effect of elastomer PDMS as a healing layer was examined (sample 3). Gas permeation measurements were conducted for the pre-combustion gaseous mixture ($H_2/CO_2$, 25/25 cm$^3$ (STP) min$^{-1}$) under a feed pressure of 4 bar and a temperature of 200°C for up to 51 days of cumulative operation. See electronic supplementary material, tables S2–S4, for detailed values, as well as figures 5–7 for the results of samples 1, 2 and 3, respectively. Figure 8 shows the entire set of

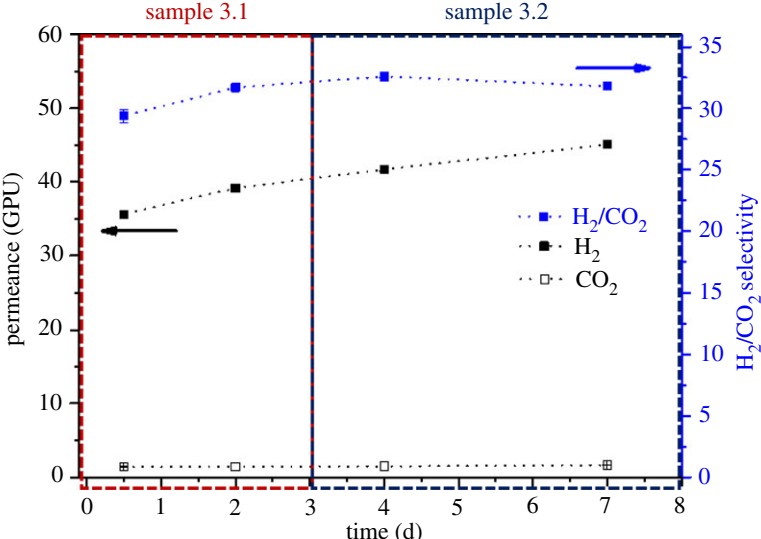

**Figure 7.** Long-term performance of HFs healed with PDMS at 200°C and 4 bar feed pressure from samples 3.1 and 3.2 considering the $CO_2$ and $H_2$ permeances as well as $H_2/CO_2$ selectivity. No sweep gas used for all the measurements.

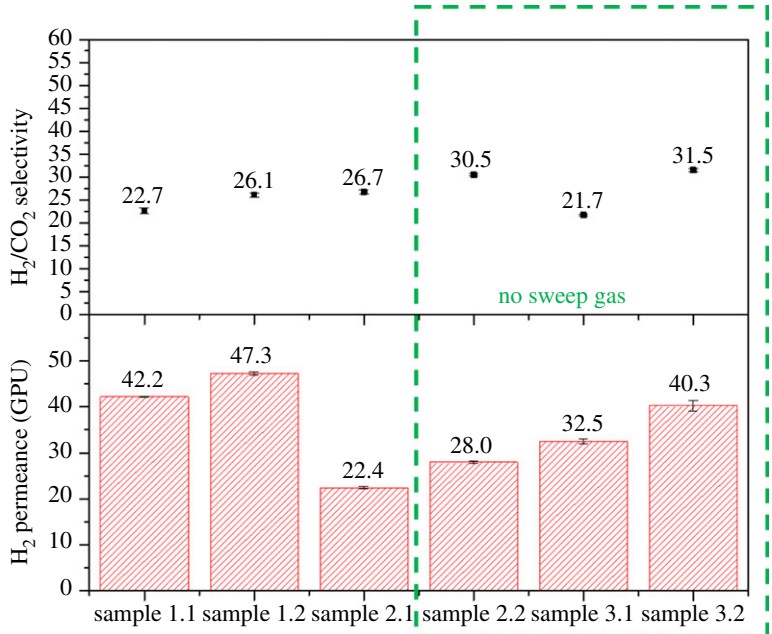

**Figure 8.** Experimental results of long-term experiments.

experiments conducted in this work. At the high operation temperature, where $CO_2$ solubility becomes unimportant, the separation mechanism is due to the preferential diffusion of the smallest $H_2$ molecule with kinetic diameter of 0.29 nm compared to that of 0.33 nm for $CO_2$.

Considering long-term experiments, it can be seen from figures 5–7 that the $H_2$ permeance as well as $H_2/CO_2$ selectivity increased after a stabilization procedure in all the measured samples; this stabilization mostly happened during the first week of operation what can be related to a final conditioning of the membrane under the operation conditions. The performance of all the mixed matrix HF membranes was stable up to 51 days without any significant fluctuations which ensures that the membranes were defect-free and that the nano-sized ZIF-8 was homogeneously distributed across the fibres. For all the membrane samples studied, the average $H_2$ permeances are in the 22–47 GPU range corresponding to 22–32 $H_2/CO_2$ average selectivity values.

Figures 5–7 show that as a function of time in the case of membrane samples from 1.1 to 3.2 both the $H_2$ permeance and the $H_2/CO_2$ selectivity values show some stabilization of the performance after about

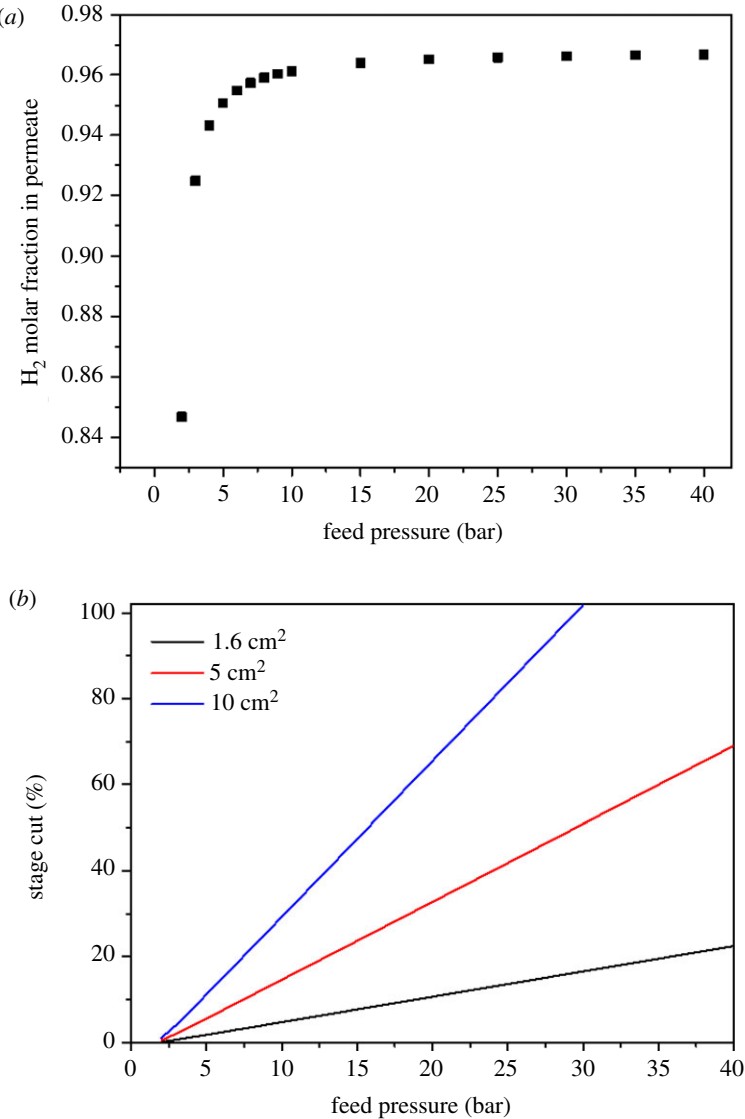

**Figure 9.** $H_2$ molar fraction in the permeate as a function of feed pressure (a) and stage cut as a function of feed pressure and membrane area (b). The separation performance of these estimations corresponds to sample 3.2.

40 days of continuous operation. Figure 8 shows the average values of $H_2$ permeance as well as $H_2/CO_2$ selectivity for all measurements conducted in this work. It can be seen that in the case of membrane samples 2.1 and 2.2 the $H_2$ permeance decreased significantly in comparison to samples 1.1 and 1.2, while the $H_2/CO_2$ selectivity increased slightly. It might be connected with slightly different physical parameters of the fibre resulting from the fabrication procedure, such as: fluctuation in the thickness of the fibre, slightly different ZIF-8 distribution along the fibre matrix regarding one fibre or another or presence of some micro-defects. It is also evident that the introduction or removal of a sweep gas in the experimental system did not show any significant effect on the final performance of the membrane, the absence of sweep gas being particularly interesting from an industrial point of view where the use of sweep gas would be prohibited. It is worth mentioning that the suppression of the sweep gas (typical of laboratory testing) allows one to advance in the demonstration of the suitability of these membranes to operate under realistic conditions.

The HFs of sample 3 (figure 7) were coated with a very thin PDMS layer to study the effect of the protective layer on the $H_2$ permeance and $H_2/CO_2$ selectivity. It is evident that in the case of sample 3.2 the obtained selectivity (31.5, corresponding to a $H_2$ permeance of 40.3 GPU) is the highest among other samples which may suggest that some micro voids or defects were healed by the thin PDMS layer.

**Table 4.** Comparison of PBI + ZIF-8 hollow fibres from the literature.

| membrane | MOF loading (wt%) | T (°C) | H₂/CO₂ feed ratio | pressure (bar) | permeance H₂ (GPU) | H₂/CO₂ selectivity | ref |
|---|---|---|---|---|---|---|---|
| ZIF-8 +PBI/ Matrimid® | 10 | 180 | 50 : 50 | 7 | 64.5 | 12.3 | [21] |
| ZIF-8 +PBI/ Matrimid® | 33 | 180 | 50 : 50 | 7 | 202 | 7.7 | [21] |
| ZIF-8+PBI | 10 | 150 | 50 : 50 | 7 | 107 | 16.1 | [10] |
| 30/70 (w/w) ZIF-8/PBI | 31 | 35 | 50 : 50 | 2 | 39 (Barrer) | 6.8 | [22] |
| 60/40 (w/w) ZIF-8/PBI | 20 | 35 | 50 : 50 | 2 | 670 (Barrer) | 2.8 | [22] |
| ZIF-8+PBI | 10 | 200 | 50 : 50 | 4 | 22–47 | 22–32 | this work |

Finally, the results corresponding to sample 3.2 (1.6 cm² of membrane area, 40.3 GPU of H₂ permeance and a H₂/CO₂ separation factor of 31.5) were used to estimate the H₂ molar fraction in the permeate as a function of feed pressure (figure 9a). Moreover, stage cut as a function of feed pressure (figure 9b) and for several values of membrane area was calculated as well. The H₂ molar fraction in the permeate was calculated using the following equation [20]:

$$(\alpha - 1)y_{H2}^2 + (1 - \alpha - \phi - (\alpha - 1)\phi x_{H2})y_{H2} + \alpha\phi x_{H2} = 0, \tag{3.1}$$

where $x_{H2}$ and $y_{H2}$ are the hydrogen molar fractions in feed and permeate, respectively, $\alpha$ is the separation factor and $\phi$ is the feed/permeate pressure ratio.

Even if the separation experiments were carried out at 4 bar pressure feed, figure 9a shows that there is a significant increase in the H₂ molar fraction (0.961) in permeate up to approximately 10 bar of feed pressure, obtaining a moderate improvement of such value (0.967) at 40 bar. In addition, at the feed pressure of 10 bar, the stage cut, i.e. the ratio between permeate and feed flows, is equal to 4.8%, 15% and 29% for membrane areas of 1.6 cm² (the one used in the experiments presented in this work), 5 cm² and 10 cm², respectively. This means that for the current feed flow (50 cm³ min⁻¹) it is possible to increase the stage cut to the desired value (to be determined from an economical optimization, out of the scope of this work) through the modification of the membrane module size (visualized in terms of membrane area).

Summarizing, table 4 shows a comparison of different research found in the literature of PBI hollow fibre performance with ZIF-8 as a filler prepared by spinning. The temperature and pressure conditions change as well as the experimental systems; however, the results obtained in this work are among the best in the literature.

# 4. Conclusion

This article shows the fabrication and long-term testing procedure of PBI based mixed matrix HF membranes with ZIF-8 as a filler. The purpose of this research was to present the overall membrane fabrication with scaled-up ZIF-8 synthesis (1 kg batch) and long-term measurement that will be closer to industrial application.

Overall, it was demonstrated that PBI HF membranes with ZIF-8 can be used for a long-term experimental H₂/CO₂ separation, for more than 50 days of cumulative continuous operation, without experiencing significant fluctuations in measurements. The membranes were robust and did not leak in any part of the experimental procedure and can be potentially used for the application of pre-combustion carbon capture. Of particular interest is the demonstration of the stable and selective operation of the hollow fibre MMM at 200°C in the absence of sweep gas. These operation conditions suggest the suitability of these membranes for industrial application.

Data accessibility. The data are provided in the electronic supplementary material [23].

Authors' contributions. A.P.-C. carried out the laboratory work, participated in the membrane characterization, performed the gas permeation measurements; M.E.-B. carried out the laboratory work, fabricated the hollow fibre membranes and participated in the membrane characterization; A.D. synthesized the metal-organic framework and carried out its characterization; M.M. participated in the design of the experiments, and critically revised the manuscript; O.D. and T.J. coordinated the study and revised the manuscript; C.T. participated in the design of the study and revised the manuscript; and J.C. designed the study, coordinated the study and revised the manuscript. All authors gave final approval for publication and agreed to be held accountable for the work performed therein.

Competing interests. A.D. and T.J. work for a company with a commercial interest in MOF materials.

Funding. This project has received funding from the European Union's Horizon 2020 research and innovation programme under grant agreement no. 760944 (MEMBER project). Also, financial support from the research project PID2019-104009RB-I00/AEI/10.13039/501100011033 from the Spanish Agencia Estatal de Investigación (AEI) is gratefully acknowledged.

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
