## [Peer Review File · Royal Society Open Science]

Review History

RSOS-210660.R0 (Original submission)

Review form: Reviewer 1

Is the manuscript scientifically sound in its present form?

Yes

Are the interpretations and conclusions justified by the results?

Yes

Is the language acceptable?

Yes

Do you have any ethical concerns with this paper?

No

Have you any concerns about statistical analyses in this paper?

Yes

Recommendation?

Major revision is needed (please make suggestions in comments)

Comments to the Author(s)

In this work, the author transformed ZIF-L into ZIF-8 in a large scale and incorporated the resulting ZIF-8 crystals into PBI to prepare the ZIF-8/PBI mixed matrix hollow fiber membranes. The long-term test of H₂/CO₂ separation by these mixed matrix fibers were studied systematically. The author investigated the effects of healing coating, sweep gas and aging on resultant performance. In my opinion, this work could be accepted by this journal but it needs some revisions. Here are the detailed reasons:

1. Transforming ZIF-L to ZIF-8 is one of the points that the author highlighted. Hence, the author should supply more characterizations to demonstrate this successful strategy, such as the morphology and XRD of the sample before and after transformation.
2. In the membrane characterization section, the author pointed out that the selective layer thickness was about ca. 3-4 μ m. But the SEM images in Figure 4 could not demonstrate this because there was not a clear distinction between the selective layer and substructure.
3. The author denoted the ZIF-8 loading was 10 wt% in the fiber. However, there was an unnegotiable amount of PVP in the polymer phase, so the ZIF-8 loading of 10 wt% seems to inaccurate.
4. The contrast experiments are somewhat unreasonable. Taking the "sweep gas or not" as an example, there was not a sample that only had the variable of "no sweep gas" to contrast sample 1.1.
5. Why did H₂ permeance and H₂/CO₂ selectivity increase after aging procedure in all the measured samples?
6. The author held that "all the mixed matrix HF membranes was stable up to 51 days without any significant fluctuations which ensures that the membranes were defect-free and that the nano-sized ZIF-8 was homogenously distributed across the fibers." However, the author explained that the decreased H₂ permeance and increased H₂/CO₂ of sample 2 compared to sample 1 was attributed to the "non-homogenous ZIF-8 distribution along the fiber matrix or some micro-defects". These two descriptions are contradictory.

Review form: Reviewer 2

Is the manuscript scientifically sound in its present form?

Yes

Are the interpretations and conclusions justified by the results?

Yes

Is the language acceptable?

Yes

Do you have any ethical concerns with this paper?

No

Have you any concerns about statistical analyses in this paper?

No

Recommendation?

Accept with minor revision (please list in comments)

Comments to the Author(s)

See attachment (Appendix A).

Decision letter (RSOS-210660.R0)

Dear Dr Malankowska:

Title: Pre-combustion gas separation by ZIF-8-polybenzimidazole mixed matrix membranes in the form of hollow fibres - long term experimental study
Manuscript ID: RSOS-210660

The editor assigned to your manuscript has now received comments from reviewers. We would like you to revise your paper in accordance with the referee and Subject Editor suggestions which can be found below (not including confidential reports to the Editor). Please note this decision does not guarantee eventual acceptance.

Please submit your revised paper before 03-Jul-2021. Please note that the revision deadline will expire at 00.00am on this date. If we do not hear from you within this time then it will be assumed that the paper has been withdrawn. In exceptional circumstances, extensions may be possible if agreed with the Editorial Office in advance. We do not allow multiple rounds of revision so we urge you to make every effort to fully address all of the comments at this stage. If deemed necessary by the Editors, your manuscript will be sent back to one or more of the original reviewers for assessment. If the original reviewers are not available we may invite new reviewers.

On behalf of the Subject Editor Professor Anthony Stace and the Associate Editor Professor Chaohua Cui.

RSC Associate Editor:
Comments to the Author:
(There are no comments.)

RSC Subject Editor:
Comments to the Author:
(There are no comments.)

Reviewers' Comments to Author:

Reviewer: 1

Comments to the Author(s)

In this work, the author transformed ZIF-L into ZIF-8 in a large scale and incorporated the resulting ZIF-8 crystals into PBI to prepare the ZIF-8/PBI mixed matrix hollow fiber membranes. The long-term test of H₂/CO₂ separation by these mixed matrix fibers were studied systematically. The author investigated the effects of healing coating, sweep gas and aging on resultant performance. In my opinion, this work could be accepted by this journal but it needs some revisions. Here are the detailed reasons:

1. Transforming ZIF-L to ZIF-8 is one of the points that the author highlighted. Hence, the author should supply more characterizations to demonstrate this successful strategy, such as the morphology and XRD of the sample before and after transformation.
2. In the membrane characterization section, the author pointed out that the selective layer thickness was about ca. 3-4 μ m. But the SEM images in Figure 4 could not demonstrate this because there was not a clear distinction between the selective layer and substructure.
3. The author denoted the ZIF-8 loading was 10 wt% in the fiber. However, there was an unnegotiable amount of PVP in the polymer phase, so the ZIF-8 loading of 10 wt% seems to inaccurate.
4. The contrast experiments are somewhat unreasonable. Taking the "sweep gas or not" as an example, there was not a sample that only had the variable of "no sweep gas" to contrast sample 1.1.
5. Why did H₂ permeance and H₂/CO₂ selectivity increase after aging procedure in all the measured samples?
6. The author held that "all the mixed matrix HF membranes was stable up to 51 days without any significant fluctuations which ensures that the membranes were defect-free and that the nano-sized ZIF-8 was homogenously distributed across the fibers." However, the author explained that the decreased H₂ permeance and increased H₂/CO₂ of sample 2 compared to sample 1 was attributed to the "non-homogenous ZIF-8 distribution along the fiber matrix or some micro-defects". These two descriptions are contradictory.

Reviewer: 2
Comments to the Author(s)
see attachment.

Author's Response to Decision Letter for (RSOS-210660.R0)

See Appendix B.

RSOS-210660.R1 (Revision)

Review form: Reviewer 1

Is the manuscript scientifically sound in its present form?

Yes

Are the interpretations and conclusions justified by the results?

Yes

Is the language acceptable?

Yes

Do you have any ethical concerns with this paper?

No

Have you any concerns about statistical analyses in this paper?

No

Recommendation?

Accept as is

Comments to the Author(s)

The authors have well addressed the reviewers' comments.

Review form: Reviewer 2

Is the manuscript scientifically sound in its present form?

Yes

Are the interpretations and conclusions justified by the results?

Yes

Is the language acceptable?

Yes

Do you have any ethical concerns with this paper?

No

Have you any concerns about statistical analyses in this paper?

No

Recommendation?

Accept as is

Comments to the Author(s)

No comments.

Decision letter (RSOS-210660.R1)

Dear Dr Malankowska:

Title: Pre-combustion gas separation by ZIF-8-polybenzimidazole mixed matrix membranes in the form of hollow fibres – long term experimental study
Manuscript ID: RSOS-210660.R1

It is a pleasure to accept your manuscript in its current form for publication in Royal Society Open Science. The chemistry content of Royal Society Open Science is published in collaboration with the Royal Society of Chemistry.

On behalf of the Subject Editor Professor Anthony Stace and the Associate Editor Professor Chaohua Cui.

RSC Associate Editor:
Comments to the Author:
(There are no comments.)

RSC Subject Editor:
Comments to the Author:
(There are no comments.)

Reviewer(s)' Comments to Author:
Reviewer: 1
Comments to the Author(s)
The authors have well addressed the reviewers' comments.

Reviewer: 2
Comments to the Author(s)
No comments.

Appendix A

Review RSOS-210660, “Pre-combustion gas separation by ZIF-8-polybenzimidazole mixed matrix membranes in the form of hollow fibres – long term experimental study”

In this manuscript, the H₂/CO₂ separation properties of PBI hollow fibres, prepared as mixed matrix membranes with ZIF-8 as filler was studied. ZIF-8 was synthesised upon up-scaling the MOF synthesis into a 1 kg batch. PBI HF membranes with ZIF-8 shown good separation performance in long-term H₂/CO₂ separation experiments. From my point of view, this paper can be published in *Royal Society Open Science* after minor revision.

Detailed comments are listed as follows:

1. What is the difference between this research and previous studies in Reference [10]? (Separation and Purification Technology, 237, 116347) Both PBI and ZIF-8 were used for the separation of H₂/CO₂.

2. In this research, 1 kg batch ZIF-8 was mentioned in several parts, is this one of the main innovations here? What is the difference of ZIF-8 synthesis process between this work and the others in references?

3. In Page 14, line 41, the author mentioned “samples 2.1 and 2.2 the H₂ permeance decreased significantly in comparison to samples 1.1 and 1.2,” and attributed this to the slightly different physical parameters of the fibre. Is this proved by any instrument characterizations? More discussion and characterization should be added here.

4. In Table 4, more references should be added.

Dr. Magdalena Malankowska
Chemical & Environmental Engineering
Department
University of Zaragoza
Institute of Nanoscience of Aragon
c/Mariano Esquillor, s/n 50018 Zaragoza **SPAIN**
PHONE: 34 669 984847
E-mail: magnal@unizar.es

Zaragoza, July 14th, 2021

Dear Editor,

Please find enclosed the revised manuscript **RSOS-210660** “Pre-combustion gas separation by ZIF-8-polybenzimidazole mixed matrix membranes in the form of hollow fibres – long term experimental study” by Adelaida Perea-Cachero, Miren Etxeberría-Benavides, Oana David, Adam Deacon, Timothy Johnson, Magdalena Malankowska, Carlos Téllez and Joaquín Coronas for consideration for publication in Royal Society Open Science as a paper.

In the next pages (those corresponding to the letter to referees) you will find our answers to the point-by-point comments made by the reviewers. We highlighted in yellow the text changed or added in the manuscript. Moreover, we also submitted a pdf version of our article (submitted to another journal) as a response to the reviewers questions.

We appreciate your efficient management concerning our work.

Thanks in advance. I look forward to hearing from you soon.

Yours sincerely,

Magdalena Malankowska

Reviewer 1

General comments: In this work, the author transformed ZIF-L into ZIF-8 in a large scale and incorporated the resulting ZIF-8 crystals into PBI to prepare the ZIF-8/PBI mixed matrix hollow fiber membranes. The long-term test of H₂/CO₂ separation by these mixed matrix fibers were studied systematically. The author investigated the effects of healing coating, sweep gas and aging on resultant performance. In my opinion, this work could be accepted by this journal but it needs some revisions. Here are the detailed reasons:

Answer: We thank the reviewer for their time revising our manuscript and for all the useful comments and remarks. Below, we provide the point-by-point replies. All the modifications in the manuscript are marked in yellow.

Question 1: Transforming ZIF-L to ZIF-8 is one of the points that the author highlighted. Hence, the author should supply more characterizations to demonstrate this successful strategy, such as the morphology and XRD of the sample before and after transformation.

Answer: We thank the reviewer for this comment. In fact, we have just submitted new manuscript to Nature Communications (ref: NCOMMS-21-27332) titled: “Understanding the ZIF-L to ZIF-8 transformation – from fundamentals to fully costed kilogram-scale production”, where all the characterization of both ZIFs are presented and discussed. We attach a pdf version of this manuscript.

Question 2: In the membrane characterization section, the author pointed out that the selective layer thickness was about ca.3-4 μm. But the SEM images in Figure 4 could not demonstrate this because there was not a clear distinction between the selective layer and substructure.

Answer: We thank the reviewer for this comment. We believe that Figure 4 B shows the selective layer clearly. Especially in the area B1 the distinctive layer of the selective part is visible. We added additional arrows to Figure 4 B to emphasize the position of the selective layer (see figure below).

Question 3: The author denoted the ZIF-8 loading was 10 wt% in the fiber. However, there was an unnegotiable amount of PVP in the polymer phase, so the ZIF-8 loading of 10 wt% seems to inaccurate.

Answer: We thank the reviewer for pointing this out. ZIF-8 loading has been calculated considering just the PBI as polymer phase. ZIF-8 loading calculation formula described in page 5 has been corrected for clarification: “wt.ZIF-8 / (wt.ZIF-8+wt.PBI) x 100”.

Question 4: The contrast experiments are somewhat unreasonable. Taking the "sweep gas or not" as an example, there was not a sample that only had the variable of "no sweep gas" to contrast sample 1.1.

Answer: We thank the reviewer for pointing this out. Figure 1 shows all the variables and all the measured samples. “No sweep gas” was also used in samples 2.2 (also without PDMS coating) and in samples 3.1 and 3.3 (with PDMS as an additional layer). The main reason of these experiments was to imitate the “real case scenario” as much as possible, hence, for example lack of sweep gas in some samples. A new sentence has been added

to pages 14-15 highlight this condition: “It is worth mentioning that the suppression of the sweep gas (typical of laboratory testing) allows one to advance in the demonstration of the suitability of these membranes to operate under realistic conditions.”

Question 5: Why did H₂ permeance and H₂/CO₂ selectivity increase after aging procedure in all the measured samples?

Answer: We thank the reviewer for this remark. Physical aging usually produces a reduction in gas permeability of glassy polymers and high fractional free volume polymers. However, it has been demonstrated that physical aging depends on the thickness of the polymeric membrane, especially when membrane thickness is less than 1 μm¹. Physical aging rate increases as membrane thickness decreases and the effect is higher when ultrathin films are considered (less than 400 nm). Membrane skin layer thickness in this study was around 3-4 μm, higher than 1 μm. Consequently, the physical aging on these PBI hollow fibres may not have a strong influence (in the time scale) and the changes observed in the membrane operation could be due to its slow stabilization. A new sentence has been added to pages 13-14 as follows:

“Considering long-term experiments, it can be seen from Figures 5-7 that the H₂ permeance as well as H₂/CO₂ selectivity increased after a stabilization procedure in all the measured samples; this stabilization mostly happened during the first week of operation what can be related to a final conditioning of the membrane under the operation conditions.”

Question 6: The author held that "all the mixed matrix HF membranes was stable up to 51 days without any significant fluctuations which ensures that the membranes were defect-free and that the nano-sized ZIF-8 was homogeneously distributed across the fibers." However, the author explained that the decreased H₂ permeance and increased H₂/CO₂ of sample 2 compared to sample 1 was attributed to the "non-homogeneous ZIF-8 distribution along the fiber matrix or some micro-defects". These two descriptions are contradictory.

Answer: We thank the reviewer for this remark. Every sample of the HF module that was tested was stable up to 51 days. However, the small differences among the HF membranes performance could be due to the differences in ZIF-8 distribution in one fibre in comparison to another. It is now clarified in the text (page 14, marked yellow):

“It might be connected with slightly different physical parameters of the fibre resulting from the fabrication procedure, such as: fluctuation in the thickness of the fibre, slightly different ZIF-8 distribution along the fibre matrix regarding one fibre or another or presence of some micro-defects.”

Reviewer 2

General comments: In this manuscript, the H₂/CO₂ separation properties of PBI hollow fibres, prepared as mixed matrix membranes with ZIF-8 as filler was studied. ZIF-8 was synthesised upon up-scaling the MOF synthesis into a 1 kg batch. PBI HF membranes with ZIF-8 shown good separation performance in long-term H₂/CO₂ separation experiments. From my point of view, this paper can be published in *Royal Society Open Science* after minor revision. Detailed comments are listed as follows:

Answer: We thank the reviewer for their time revising our manuscript and for all the useful comments and remarks. Below, we provide the point-by-point replies. All the modifications in the manuscript are marked in yellow.

Question 1: What is the difference between this research and previous studies in Reference [10]? (Separation and Purification Technology, 237, 116347) Both PBI and ZIF-8 were used for the separation of H₂/CO₂.

Answer: We thank the reviewer for this remark. The work presented in this manuscript is the continuation of the work from the article from Ref [10]. The main differences between these two works are: 1) long-term testing for more than 50 days, 2) study of the effect of different parameters on the final performance (sweep gas use, PDMS coating layer etc.), 3) ZIF-8 used from the 1 kg batch ecological synthesis. This has been highlighted at the end of the Introduction section (page 2) as follows:

“Finally, this works continues our previous report on this type of application with important novelties dealing with the use of ZIF-8 for the MMM coming from a green methodology, the long-term gas separation testing for more than 50 days and the study on the membrane operation under more realistic conditions (e.g. without the use of sweep gas).”

Question 2: In this research, 1 kg batch ZIF-8 was mentioned in several parts, is this one of the main innovations here? What is the difference of ZIF-8 synthesis process between this work and the others in references?

Answer: We thank the reviewer for this comment. See our answer to comment #1 by Referee 1.

Question 3: In Page 14, line 41, the author mentioned “samples 2.1 and 2.2 the H₂ permeance decreased significantly in comparison to samples 1.1 and 1.2,” and attributed this to the slightly different physical parameters of the fibre. Is this proved by any instrument characterizations? More discussion and characterization should be added here.

Answer: The attribution of the differences in the results of samples 2.1 and 2.2. to 1.1 and 1.2 is our conclusion; see also our answer to comment #6 by Referee 1. In any event, it was not supported by any characterization since the HFs are long and it was not possible to measure each section under the microscope.

Question 4: In Table 4, more references should be added.

Answer: More references are added to the table (see below).

Table 4 Comparison of PBI+ZIF-8 hollow fibres from the literature

Membrane	MOF loading (wt.%)	T (°C)	H ₂ /CO ₂ feed ratio	Pressure (bar)	Permeance H ₂ (GPU)	H ₂ /CO ₂ selectivity	Ref
ZIF-8 +PBI/Matrimid®	10	180	50:50	7	64.5	12.3	[21]
ZIF-8 +PBI/Matrimid®	33	180	50:50	7	202	7.7	[21]
ZIF-8+PBI	10	150	50:50	7	107	16.1	[10]
30/70 (w/w) ZIF-8/PBI	31	35	50:50	2	39 (Barrer)	6.8	[22]
60/40 (w/w) ZIF-8/PBI	20	35	50:50	2	670 (Barrer)	2.8	[22]
ZIF-8+PBI	10	200	50:50	4	22-47	22-32	This work

References:

1. D. E. Sanders, Z. P. Smith, R. L. Guo, L. M. Robeson, J. E. McGrath, D. R. Paul and B. D. Freeman, *Polymer*, 2013, **54**, 4729-4761.